# A cross sectional survey of knowledge, attitude and practices related to the use of insecticides among farmers in industrial triangle of Punjab, Pakistan

Bakhtawer, Sumera Afsheen*

Department of Zoology, University of Gujrat, Hafiz Hayat Campus, Gujrat, Punjab, Pakistan

* sumera.afsheen@uog.edu.pk

## Abstract

Pesticides in Pakistan are abundantly utilized for pest control in agriculture sector. The over and unsafe use of insecticides plus poor handling leads to the development of resistance, outbreak of secondary pests and hazardous impact on environment. The present study was aimed to access the current knowledge, attitude and common practices of farmers about the use of insecticides against pest in industrial triangle of Province Punjab, Pakistan. This study was conducted during October 2019 to February 2020. In this study farmers (n = 300) took part from three localities of Province Punjab (Gujrat, Gujranwala and Sialkot). Farmers were interviewed using a questionnaire to collect data about the knowledge of pest control by use of chemical method, biological method and combination of both to eradicate the pests. The result shows almost all (93%) farmers were male and they did not know about the insecticides mode of action and its chemical composition. They do not have any knowledge about the biological control of pests and did not get any assistance or help from Agriculture Extension Officer. They even did not properly dispose off the empty containers of insecticides. Statistical analysis reveals that lack of education and awareness about biological control of pest depicts development of resistance and outbreak of secondary pest including health hazards and environmental pollution. Poor understanding about pests, abundant use of insecticides, incorrect perception about application of insecticides and negligence regarding biological control shows that there is need to initiate public awareness programs to ensure the application of integrated pest management (IPM) and sustainable agriculture.

## Introduction

Being an agricultural country, economy of Pakistan is based on agriculture sector. There is about 1.8 billion people round the globe that deal with pesticides to increase the yield of crops and protect them from pest attack. In Pakistan, 42.5 percent people in rural areas are involved in agricultural activities. Chemical control of pests in agriculture sector is the major prominent method to maximize productivity of crops.

**Data Availability Statement:** All relevant data are in the paper and its Supporting Information files.

**Funding:** The authors received no specific funding for this work.

**Competing interests:** The authors have declared that no competing interests exist.

Various chemicals including Organophosphate, Neonicotinoids, Carbamates and Pyrethroids are abundantly used. There were 108 types of insecticides, 39 types of herbicides, 6 types of rodenticides and 5 types of aracicides reported which is be used in Pakistan for different pests of crops. The use of pesticides have increased by 9% or more per hectare in most developing countries including Pakistan.

It has been observed that farmers mostly rely on chemicals to protect their crops from pest's attacks without knowing the harmful consequences of insecticides. This practice leads not only to development of resistance in pests but also reduce the number of useful insects, outbreak of secondary pests, health issues and environmental pollution [1].

In Province Punjab variety of crops are cultivated including wheat, rice, cotton, sugarcane, fruits and vegetables. In these crops field sap sucking insects cause damage to crops at all stages of crop development. Farmers of this region rely on insecticidal control of pests [2]. These farmers did not have any awareness about the chemical nature of insecticides [3]. They also do not have any assistance or training from agricultural extension sector regarding application of insecticides [4]. Due to lack of knowledge about harmful results of pesticides farmers did not pay attention about taking precautionary measures. Improper disposal of empty containers that contain surplus enough quantity of pesticides leads to the pollution as well as resistance development in non target species.

For sustainable agriculture there is need to reduce the use of insecticides and control the pest attack by other means. It is very necessary to adopt the biological control method along with chemical control in order to bring sustainability. For this purpose, public awareness is very important to give general knowledge to farmers about the use of pesticides. Integration of chemical control with biological control is the basic part of IPM but unfortunately it has been rarely applied [5]. Use of natural predators can be economical to farmers as well as it is ecofriendly.

Keeping in view all the factors current study was designed to access the knowledge of farmers about pesticides application and alternative methods to control pests. This will help the agriculture department to design a comprehensive plan to educate farmers as well as built up an effective IPM strategy to control pests attack, secure food and bring sustainability.

## Materials and methods

### Study area

A cross sectional study was conducted during October 2019 to February 2020 by using questionnaire based interview of farmers. In this survey face to face conversation was done with farmers. Firstly, in local language aim of study was described, later on after their willingness questions were asked verbally and their response was noted on questionnaire. This survey was carried out in industrial triangle including Gujrat, Gujranwala and Sialkot. These sites were selected on the basis of operational convenience and industrial importance of this region. Due to availability of water and suitability of climatic conditions various crops like wheat, rice, fruits and vegetables are cultivated in this region. Farmers of this region use insecticides abundantly to protect their crops from pests. The demographic characteristics of this region are given in Table 1.

From each site six localities were further selected randomly. From Gujrat, six localities include Jalal pur jattan (JPJ), Karianwala (KN), Dolat Nagar (DN), Wazirabad (w), Lalamusa (LM) and Kharian (KH). From Gujranwala, ferozawala (FW), Uggochak (UG), Veropal Chatha (VC), Ali pur Chatha (AC), Qila miyan singh (QS) and Kot Ladha (KL) were selected for data collection. Daska (D), sambrial (S1), Sahowal (S2), Majra Kalan (MK), Dera sanda (SD) and Sahowala (S3) from Sialkot site were included in survey study. Detail of study area is represented in Map 1.

**Table 1. Demographic characteristic of study area source: Pakistan bureau of statistics 2017.**

| Province | District | Area Sq km | Population | | |
|----------|----------|------------|------------|------|------|
| | | | Number | Rural % | Urban % |
| Punjab | Gujranwala | 3622 | 5014196 | 41.18 | 58.81 |
| | Gujrat | 3192 | 2756110 | 96.98 | 30.02 |
| | Sialkot | 3016 | 3893672 | 70.64 | 29.36 |

In first step questionnaire was prepared and survey was conducted from a subsample of 20 farmers who were reluctant to participate in study. On the basis of results of pre-test, this survey was further modified to reduce the deficiencies felt at that time. In second step a comprehensive survey was conducted among 300 respondents of District Gujrat, Gujranwala and Sialkot of Punjab province.

### Interview procedure

Each village was visited for 4–5 days and 10–15 on average respondents were interviewed per day via face to face conversation. Before interview main objectives were explained clearly to respondents. Later on, with the consent of the respondent, the data was recorded on questionnaire by verbally asking the questions because majority of respondents were not able to read and understand questionnaire.

A structured questionnaire having open ended and close ended questions was used to collect information by using Urdu and Punjabi Language. The questionnaire consists of three sections. First part is related to sociodemographic characters of respondents such as Gender, marital status, age, education level, farm area, irrigation method, farming experience and working hours they spend in their crops. Second part is related to perceptions like which insecticide is better and they prefer to use more, which crop and which pest most frequently targeted, how they prepare the spray dose for targeted pests, they have any idea about alternate pest control method like IPM, Biological agent or natural enemies and third part related to the attitude and practices of respondents like how they protect themselves during spray, they use protective equipment or not and they acquired any skill or training from any department or agency.

### Ethics statement

This study was approved by all statuary forums of the University of Gujrat, Gujrat, Pakistan including the Departmental Review and Research Committee (DRRC), Advanced Study and Research Board (ASRB), Board of Faculty (BoF) and Academic Council.

## Results

In total, 300 respondents from three Districts participated in this study. Random sampling was done. Most of the respondents were involved in cultivation of wheat, rice, millet, sorghum and vegetables in this region. The response rate from respondents was 100% which makes this study more interesting.

### Demographic characteristics of respondents

The majority of respondents were male (i.e.93.66%) while only 6.34% women were involved in farming. Education is a very important variable for the assessment of farmer's knowledge and practices. It was found that 34% respondents were illiterate, 16% had primary level, 6% had middle level education, 11.33% of respondents had education level up to higher secondary

**Map of the studied area showing the locations of the three District of Punjab, Pakistan**

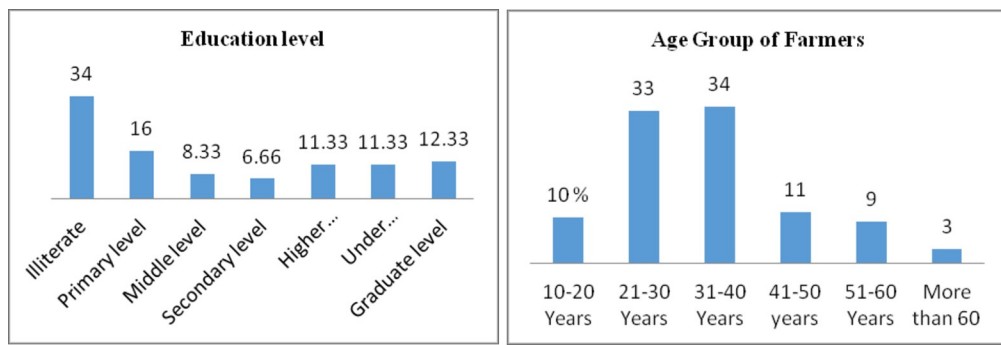

**Map 1. Map indicating current study area (Map created by using GIS-ARc).**

school and only 12.33% were graduated. Fig 1(A) represents the detail of the education level of respondents. Age group of the respondents ranged from 20–60 years with an average of 31–40 (34%) years. Detail of age in years was represented in Fig 1(B).

## Description of respondents

Most of the respondents have 1-hectare area for the cultivation of crops. This data indicate that these farmers had small farming units (Fig 2).

**Fig 1. (a). Education level.** Education level of the farmers represented in Fig 1 (a). Percentage of farmers was given on Y-axis while their education level was placed on X-axis. **(b). Age Group of Farmers.** Age group of farmers was represented on x-axis and their percentage was given on Y-axis.

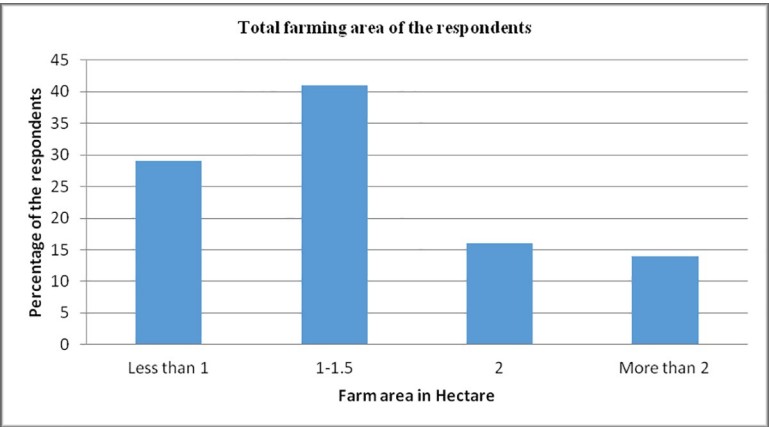

**Fig 2. Total farming area of the respondents.** Farming area of respondents in hectare represented in Fig 2. The area occupied by respondents in hectare was placed at x-axis while Y-axis represents the percentage of the respondents. Most of the respondents has farm area in the range of 1–1.5 hectare followed by less than 1 hectare however respondents having 2 or more than 2 hectares were having same trend.

Most common irrigation method adopted by the respondents was tube well (i.e.72%) followed by the drip method (i.e.18%) while 3% was reported to utilize the canal system for irrigation purpose as shown Fig 3. Most of the respondents had 7–9 years of farming experience is presented in Fig 4.

38% respondents spend 4–6 hours for farming activities daily. Only 2% reported to do work more than 12 hours.

## Knowledge of respondents regarding pesticides

Majority of respondents deals with acetamprid, cypermethrim, lambda cyhalothrin and dimethoate belonging to the organophosphate and pyrethoids group of insecticides (WHO category II). Pest infestation was reported by the respondents including Smuts, Aphids,

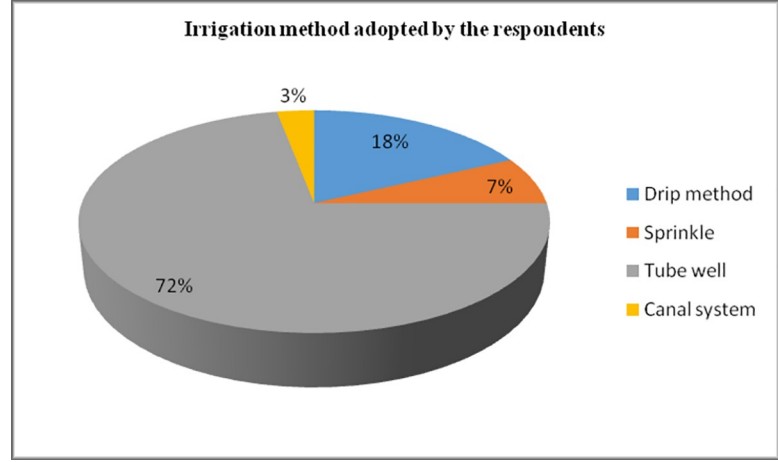

**Fig 3. Irrigation method adopted by the respondents.** Irrigation method-This pie chart represent the irrigation method adopted by the respondents.72% of the respondents use tube well system for irrigation purpose, 18% use drip method, 7% use sprinkle method and 3% use canal system for irrigation of their crops.

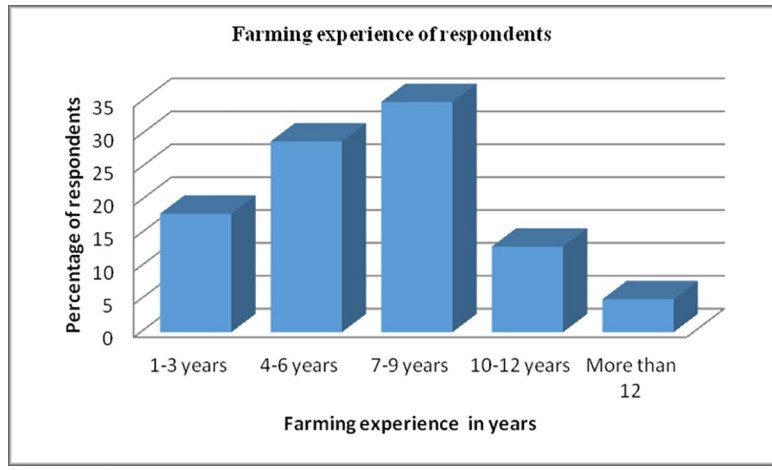

**Fig 4. Farming experience of respondents.** Farming experience- this graph represents the farming experience of the respondents in years. On X-axis faring experience in years was mentioned while on Y-axis percentage of the respondents was mentioned. This data revealed that most of the respondents have 7–9 years of farming experience.

Whiteflies, Jassids, Mildew, Mealybugs and Termites in this area. For the control of pest attack 46% farmers used to spray their crops three times, 32% sprayed four times and 6% were reported to spray their crops five times. These sprays were used during the early stage of pest growth (67.33%). Only 5.66% were reported to use spray at harvesting stage of crops. About 71.33% of farmers spray their crops by arial method, 15.33% used synergist and only 13.33% used mixture with water. Label instruction for preparing dose is very crucial step to properly use insecticides against pests. 42% of respondents understand the label instruction during preparation for spray. 22% of respondents were able to make proper dose whereas 15% of the respondents follow the schedule for spray and 21% were found to maintain empty containers of insecticides. When the respondents were asked about the alternative of synthetic insecticides,63% did not know while 37% were aware of it. 68.6% did not have knowledge about IPM. 65.3% of respondents did not know about biological control method of pests. Detailed information about knowledge of respondents about insecticides usage and alternative pest control methods is given in Table 2.

## Common practices of respondents

Chemical control of pest (i.e. 55.33%) was common practice followed by the cultural control methods (i.e. 32. 33%). Cultural control refers to the manipulation of old practices like date management with respect to pest outbreak etc. Only a small fraction of respondents have adopted IPM (i.e. 5.33%) and biological control method (7%) in their farming routines. Interestingly, 50.33% respondents burry the empty containers and 14% of respondents burnt them while 31.67% throw them in trash without any processing while 3.67% of respondents go for Government collection centers for disposal of empty containers of insecticides. As far as precautionary measures are concerned, hand gloves 44% and facemask 41% or covering the face with cloth are most adopted precautionary measures. Use of glasses, overall, respirator and long shoes were mostly neglected measures among respondents during the use of insecticides. For the proper use of insecticides, training is very important but unfortunately only 7% have some acquired skill and 12% have training about the use of insecticides while most of them get name of products and use them for pests just by asking from agriculture officer. Detail of common practices of respondents is given in Table 3.

**Table 2. Knowledge of respondents in three localities of province Punjab Pakistan (n = 300).**

| Knowledge of farmers | | | | |
|---|---|---|---|---|
| Insecticides used | | Acetamiprid | | |
| | | Cypermethrin | | |
| | | Lambda cyhalothrin | | |
| | | Dimethoate | | |
| Characters | | | Number | Percentage |
| Crops | | Wheat | 93 | 31 |
| | | Rice | 69 | 23 |
| | | Sugarcane | 3 | 1 |
| | | Millet | 6 | 2 |
| | | Sorghum | 12 | 4 |
| | | Vegetables | 117 | 39 |
| Pests | Scientific belonging | Common name | | |
| | *Ustilaginaceae* | Smuts | 30 | 10 |
| | *Aphidoidea* | Aphids | 147 | 49 |
| | *Aleyrodidae* | Whiteflies | 36 | 12 |
| | *Amrasca biguttula* | Jassids | 42 | 14 |
| | *Golovinomyces orontii* | Mildew | 30 | 10 |
| | *Pseudococcidae* | Mealybugs | 9 | 3 |
| | *Isoptera* | Termites | 6 | 2 |
| Frequency of spray (per Month) | | Two times | 44 | 14.66 |
| | | Three time | 139 | 46.33 |
| | | Four time | 97 | 32.33 |
| | | Five time | 20 | 6.66 |
| Stage of crop | | Early stage | 202 | 67.33 |
| | | Grown stage | 56 | 18.67 |
| | | Mature stage | 25 | 8.33 |
| | | Harvesting stage | 17 | 5.67 |
| Mode of application | | Arial | 214 | 71.33 |
| | | Mix with water | 40 | 13.33 |
| | | Synergist | 46 | 15.33 |
| Label instructions | | Follow | 126 | 42 |
| | | Proper dose | 66 | 22 |
| | | Schedule | 45 | 15 |
| | | Maintain containers | 63 | 21 |
| Alternative synthetic insecticides | | Yes | 111 | 37 |
| | | No | 189 | 63 |
| In case yes | | Crop repetition | 3.33 | 3 |
| | | Organic farming | 1.11 | 1 |
| | | Crop rotation | 96.57 | 87 |
| | | Crop mixtures | 7.77 | 7 |
| Knowledge about IPM | | Yes | 94 | 31.33 |
| | | No | 206 | 68.67 |
| Biological control | | Yes | 104 | 34.67 |
| | | No | 196 | 65.33 |
| Knowledge about natural enemies | | Yes | 99 | 33 |
| | | No | 201 | 67 |

**Table 3. Common practices of respondents in three localities of province Punjab Pakistan (n = 300).**

|  | Character | Number | Percentage |
|---|---|---|---|
| Crop practices methods | Cultural | 97 | 32.33 |
|  | Chemical | 166 | 55.33 |
|  | Biological | 21 | 7 |
|  | IPM | 16 | 5.33 |
| Practices for Disposal of insecticides containers | Govt collection | 11 | 3.67 |
|  | Burry | 151 | 50.33 |
|  | Burnt | 43 | 14.33 |
|  | Throw in trash | 95 | 31.67 |
| Farmer training /skill | Pesticides application | 36 | 12 |
|  | Agriculture officer guidance | 243 | 80 |
|  | Use acquired skills | 21 | 7 |
| Practice to use Protective measures | Hand gloves | 132 | 44 |
|  | Glasses | 33 | 11 |
|  | Overall | 3 | 1 |
|  | Respirator | 3 | 1 |
|  | Face mask | 123 | 41 |
|  | Boot/shoes | 6 | 2 |
| Time of spray | Morning | 132 | 44 |
|  | Noon | 16 | 5.33 |
|  | After noon | 111 | 37 |
|  | Evening | 41 | 13.67 |

Analysis revealed respondents having non formal education are at high level of risk as compared to the respondents that know about the mode of action of insecticides. There was strong correlation between education level and common practices of respondents. There was strong association between knowledge and label instructions given on products for proper dose preparation by respondents similarly time of spray, frequency of spray, use of protective equipment and disposal of empty insecticides containers has strong association with the knowledge of respondents. However knowledge about the IPM and biological control method of pests was insufficient which indicate the need of awareness campaign among farmers. Chemical control was the dominated method practiced among the respondents for the control of pests.

## Statistical analysis

The univariate regression analysis showed that all factors had significant differences with the education level of respondents (i.e. $p < 0.001$). Table 4 shows the knowledge of respondents about the alternative methods like IPM, Biological control and natural enemies is very poor however the farmers having knowledge about them are more likely to reduce the harmful impact of insecticides.

Table 4 shows the results for Education risk practices with different predictor variables. The columns are showing the categories of predictor variables and rows are showing the outcomes variables categories followed with their OR, 95% CI for OR, and their P-values. The OR = 1 presents the reference group OR. The yellow highlighted and (–) present that there was no significant association of those variables. The estimated OR of non-formal education is 2.29 times as large as in who do not hear IPM than in who heard IPM with 95%(1.46–3.36) of OR. They don't hear IPM showed high-risk levels being less likely to non-formal education (p-value = .000). The estimated OR of primary level education is 2.21 times as large as in who do not hear

**Table 4. Summary of univariate regression analysis column of the table refers to the education level of the farmers and rows refer the practices of farmers.** Shown are the odd Ration (OR) and 95% confidence interval (95% CI).

| | | Non formal | | Primary level | | Middle level | | Secondary level | | Higher secondary level | | undergraduate level | | Graduation level | |
|---|---|---|---|---|---|---|---|---|---|---|---|---|---|---|---|
| | | P-value | OR (95%CI) | P-value | OR (95%CI) | P-value | OR (95%CI) | P-value | OR (95%CI) | P-value | OR (95%CI) | P-value | OR (95%CI) | P-value | OR (95%CI) |
| **Heard IPM** | Yes | | 1 | | 1 | | 1 | | 1 | | 1 | | 1 | | 1 |
| | No | .000 | 2.29 (1.46–3.36) | .000 | 2.21 (1.46–3.36) | .011 | 2.20 (1.19–4.05) | .079 | 2.12 (.91–4.92) | .655 | 1.22 (.50–2.94) | .012 | 2.66 (1.24–5.73) | .009 | 2.77 (1.29–5.95) |
| **Biological Control** | Yes | | | | 1 | | 1 | | 1 | | 1 | | 1 | | 1 |
| | No | .000 | 2.43 (1.59–3.72) | .000 | 2.433 (1.59–3.72) | .011 | 2.200 (1.19–4.05) | .034 | 2.57 (1.07–6.15) | .187 | 1.85 (.74–4.65) | .061 | 2.00 (.97–4.12) | .004 | 3.25 (1.47–7.17) |
| **Natural Enemies** | Yes | | | | 1 | | 1 | | 1 | | 1 | | 1 | | 1 |
| | No | .025 | 1.57 (1.06–2.34) | .025 | 1.57 (1.06–2.34) | .011 | 2.20 (1.19–4.05) | .002 | 5.25 (1.80–15.29) | .374 | 1.50 (.61–3.67) | .002 | 3.71 (1.61–8.55) | .494 | 1.26 (.64–2.49) |
| **Follow label Information** | Follow | | 1 | | 1 | | 1 | | 1 | | 1 | | 1 | | 1 |
| | Proper Dose | .03 | .59 (.36–.96) | .08 | .52 (.25–1.08) | .06 | .30 (.08–1.09) | .01 | .15 (.03–.68) | .32 | .66 (.30–1.48) | .12 | .52 (.23–1.18) | .28 | .35 (.13–.89) |
| | Follow Schedule of spray | .001 | .40 (.23–.78) | .004 | .23 (.09–.63) | .206 | .50 (.17–1.46) | .014 | .15 (.03–.68) | .011 | .20 (.05–.69) | .016 | .29 (.10–.79) | .028 | .35 (.13–.89) |
| | Maintenance of Pesticide Container | .000 | .34 (.19–.61) | .08 | .52 (.25–1.08) | .46 | .70 (.26–1.08) | .02 | .23 (.06–.81) | .01 | .26 (.08–.80) | .006 | .17 (.05–.60) | .079 | .47 (.20–1.09) |
| **Irrigation method** | Drip | | 1 | | 1 | | 1 | | 1 | | 1 | | 1 | | 1 |
| | Sprinkler | .40 | .42 (.18–.96) | .05 | .22 (.04–1.02) | -- | -- | -- | -- | .73 | 1.25 (.33–4.65) | .142 | .20 (.02–1.71) | .739 | 1.25 (.33–4.65) |
| | Tube Well | .000 | 3.78 (2.28–6.28) | .000 | 3.88 (1.86–8.09) | .006 | 4.0 (1.50–10.65) | .18 | 1.85 (.74–4.65) | .001 | 6.0 (2.08–17.29) | .001 | 5.20 (1.99–13.54) | .000 | 6.5 (2.26–18.62) |
| | Canal Irrigation | .005 | .21 (.07–.61) | .054 | .22 (.04–1.02) | -- | -- | -- | -- | -- | -- | .273 | .40 (.07–2.06) | .423 | .50 (.09–2.73) |

IPM than in who heard IPM with 95%(1.46–3.36) of OR. They do not hear that IPM showed high-risk levels being less likely to primary level education (p-value = .000). Similarly estimated OR of middle-level education is 2.20 times as large as in who do not hear IPM than in who heard IPM with 95%(1.19–4.05) of OR. They do not hear IPM showed high-risk levels being less likely to middle-level education (p-value = .001). Similarly estimated OR of undergraduate level education is 2.66 times as large as in who do not hear IPM than in who heard IPM with 95%(1.24–5.73) of OR. They do not hear IPM showed high-risk levels being less likely to undergraduate level education (p-value = .012). Similarly estimated OR of graduate-level education is 2.77 times as large as in who do not hear IPM than in who heard IPM with 95%(1.29–5.95) of OR. They do not hear that IPM showed much higher risk levels being less likely to graduate-level education (p-value = .009).

Label information on the containers of the insecticides for the preparation of required dose is a very crucial factor while most of the farmers were unable to read it as many are unable to understand the terminologies and units p<0.001. There is a need to start a campaign to give awareness to farmers about it (Table 5).

**Table 5. Summary of regression analysis column of the table refers to the education level of the farmers and rows refer the practices of farmers.**

| | | Non formal education | | Primary level | | Middle level | | Secondary level | | Higher secondary level | | Under Graduation level | | Graduation level | |
|---|---|---|---|---|---|---|---|---|---|---|---|---|---|---|---|
| **Crop Practices** | **Cultural Control** | | 1 | | 1 | | 1 | | 1 | | 1 | | 1 | | 1 |
| | **Chemical Control** | .000 | 2.78 (1.72–4.48) | .090 | 1.73 (.918–3.27) | .39 | 1.44 (.61–3.37) | .166 | 2.00 (.75–5.32) | .006 | 3.228 (1.41–7.65) | .136 | 1.80 (.83–3.89) | .023 | 2.27 (1.11–4.61) |
| | **Biological Control** | .028 | .43 (.20-.91) | .019 | .26 (.08-.80) | .054 | .22 (.04-1.02) | . | . | .118 | .28 (.05-1.37) | .121 | .40 (.12–1.27) | . | . |
| | **IPM** | .003 | .26 (.10-.64) | .011 | .20 (.05-.69) | .037 | .11 (.01-.87) | .178 | .33 (.06-1.65) | .069 | .14 (.01-1.16) | .038 | .20 (.04-.91) | .022 | .09 (.01-.70) |
| **Disposal of empty containers** | **Government Collection** | | 1 | | 1 | | 1 | | 1 | | 1 | | 1 | | 1 |
| | **Burry** | .000 | 63 (8.73-454-2) | .001 | 11 (2.58–46.7) | .054 | 4.5 (.97-20.8) | .080 | 4 (.84-18.8) | .005 | 18 (2.40-134.8) | .001 | 12 (2.83-50.7) | .002 | 23 (3.1-170.3) |
| | **Burnt** | .013 | 13 (1.70-99.3) | .423 | 2 (.36-10.9) | .423 | 2 (.36-10.9) | 1 | 1 (.14-7.0) | .142 | 5 (.58-42.7) | .423 | 2 (.36-10.9) | .142 | 5 (.58-42.7) |
| | **Throw in Trash** | .001 | 26 (3.52-191.5) | .002 | 10 (2.3-42.7) | .038 | 5 (1.09-22.8) | .080 | 4 (.84-18.8) | .037 | 9 (1.14-71.03) | .423 | 2 (.36-10.91) | .050 | 8 (1-63.9) |
| **Training and Protective Measures** | **Hand Gloves** | | 1 | | 1 | | 1 | | 1 | | 1 | | 1 | | 1 |
| | **Eye Glasses** | .000 | .17 (.09-.34) | .001 | .16 (.05-.46) | .013 | .07 (.01-.58) | .017 | .08 (.01-.64) | .058 | .40 (.15–1.03) | .002 | .10 (.02-.42) | .034 | .38 (.16-.93) |
| | **Overall** | .000 | .01 (.002-.12) | . | . | .013 | .07 (.01-.58) | .017 | .08 (.01-.64) | .009 | .06 (.009-.50) | .003 | .05 (.007-.37) | . | . |
| | **Respirator** | .000 | .03 (.009-.14) | .002 | .04 (.005-.29) | . | . | . | . | .007 | .13 (.03-.58) | . | . | . | . |
| | **Face Mask** | .004 | .52 (.33-.81) | .163 | .64 (.34–1.2) | .533 | .76 (.33-1.75) | .057 | .33 (.10-1.03) | .226 | .60 (.26-1.3) | .074 | .50 (.23-1.06) | .136 | .55 (.25-1.20) |
| | **Boot/Shoes** | .000 | .05 (.01-.16) | .001 | .08 (.01-.33) | . | . | .019 | .16 (.03-.74) | . | . | .003 | .05 (.007-.37) | .003 | .11 (.02-.47) |

The regression analysis also showed that most common practice to control the pest was use of chemical abundantly by the farmers instead of alternative methods (p = 0.001).

Disposal of empty containers is a potential factor that can bring resistance in pests, pollution and outbreak of secondary pest. This practice makes the situation worse than sustainability (p = 0.001). Use of protective equipment by the farmers is neglected most (p = 0.004)- see Table 5.

## Discussion

Current study reported that most of the farmers were male (93.6%) while 6.4% were female. The study conducted by Dilek Oztas et al in Turkey reported 100% male farmers [6]. Similar work done by Oluewe and Cheke in Nigeria 93% of farmers were male. Lekei et al. also reported the same trend 93% farmers were male [7].

The mean age of the farmers were in range of 31–50 years. The mean age of farmers were 18–51 years reported by Tuna et al in a study conducted about farmer's knowledge, attitude and behavior about pesticides in Turkey [8]. The average age of farmers were 37.5 years according to Lekei et al [9]. in Tanzania. Mubashar et al. conducted study on knowledge of farmers regarding pesticides usage and biosafety in Pakistan reported average age of farmers were 30–50 years [10]. Similar trend was reported by Khuhro et al. during 2020 in northern Sindh Pakistan for 18–50 years age of farmers [11].

Education level of farmers is closely related with their practices. Educated farmers can read and understand the instructions given on insecticides about dose preparation and precautionary measures given. The current survey showed 34% farmers were illiterate, 16% had primary level education, 8.33% had middle level education while 12.33% were graduated from university and 11.33% were those who got education up to higher and undergraduate level. A study conducted by Yassin et al. (2002) on Knowledge, attitude, practices and toxic symptoms related to use of pesticides in Philistine showed 8.5% were illiterate [12]. In Northern Sindh, it was reported that 12.2% were illiterate 40.9% were primary and 27.7% had middle level education. In Lodhran and Vehari Pakistan similar trend was reported i.e. 26.4% were illiterate. These results are in accordance with Khan and Iqbal who reported that majority of farmers in Pakistan had low level of education and only 6% had university level education [13].

Experience of farmers is very important in order to acquire skills. This factor alone can help to increase yield and make farming cost effective. Current study represented that 35% farmers have 7–9 years of farming experience, 13% have 10–12 years while 29% had 4-5years of farming experience. The same trend was reported by Mubasher et al that 41% had 11–15 years 37.9% having 5–10 years of farming experience [10]. Research in Pakistan represented that sociodemographic characteristics of farmers like gender, age, education level and farming experience greatly affect the use of insecticides.

Alternative pest control method is very important component of IPM and very essential for sustainability. The result of current study shows that 63% of respondents did not have any idea about alternative methods. Among 37% that know about alternative method crop rotation only practiced among farmers. Biological control is a very effective method along with the chemical control but unfortunately due to low level of knowledge and awareness that 65.3% of respondents did not know about this. This unawareness leads to force them only rely on insecticidal control. Khan et al reported few farmers implement IPM and most did not know about non chemical method [14]. According to them cultural control and mechanical methods were easy to implement. Khuhro et al reveals that 22.72% farmers practiced cultural method, 22.72% could not identify pest and 95.45% did not use any other alternative method to control pest [11]. The work is also supported by Yassin et al farmers did not have any knowledge about biological control or natural control agent [12].

Following the instruction given on the products for the preparation of proper dose is very important to prevent the overused as well as the release of residues in environment ultimately leads to development of resistance in insects. Present study reported that only 22% farmers were able to prepare the required dose while 58% farmers were unable to understand the terminologies and language of the label instructions. There was strong association between farmer's knowledge and understanding the label information of insecticides products. It leads to the higher exposure via the increase emission rate. Dalamas and Khan reported that 73% farmers were unable to read label instruction on insecticides containers [15]. Similar trend was reported by Mubasher et al i.e. 48.2% were unable to read instructions on insecticides label [10]. Shetty et al reported major factor behind this is illiteracy and lack of awareness [16].

Failure to use of protective measures was another problem linked with the hazardous impact of insecticides on environment. A study conducted in North Greece and India reported the significant association between knowledge and use of protective measures. Clark and collegues reported the poor trend about use of protective measures among the farmers in tropics due the hot and humid climatic conditions farmers feel discomfort during use of protective measures [17].

The use of protective measure becomes an important step for the sustainable use of insecticides likewise the IPM, irrigation method, use of less toxic agents and alternative pest control methods (use of biological agents) are the key component for sustainable agriculture. In this survey it was concluded that most of the farmers neglected the use of protective gears i.e.

29.7% use face mask and 3.3% use long shoes. The use of respirator was neglected during spray which was only 1.7% reported. It is worth mentioning that 40% farmers in Iran were reported to use any protective equipment during spray reported by Hashemi et al [18].

Furthermore, improper disposal of empty containers may be another source of pesticides exposure. Mostly farmers dumped the empty insecticides containers in unsafe manners this leads to the contamination of environment due to run off, leaching and arial distribution of residues even in developing countries the practices were very common. Study in Tanzania and south Africa reported that farmers had been using empty insecticides containers for domestic purpose.

Data reveals most of respondents had no trainings from extension officer about the use of insecticides as Aldosari *et al* [19] that 79.5% of farmers did not received any training whereas 12.3%, 5.1% and 3.1% received 2 to 4 and more than four trainings respectively [19]. Similarly according to Aslam *et al* [20] farmers of Pakistan had low level of knowledge about insecticides use which attributes to the fact that training by extension officer can be an effective step to reduce risk factors [21]. According to another study conducted in Ethiopia by Negatu *et al* [21], 85% of the farmers had no training regarding application of insecticides [22], while Ibitayo [3] reported similar trend of farmers [4]. According to this study 98% of farmers did not had any training about insecticides application. In UK, a study conducted regarding risk assessment about safe use of insecticides among workers. It indicated that proper training and adoption of proper safety measures can reduce the health hazards [6].

## Conclusions

The findings of this survey demonstrate poor understanding and risky practices adopted by the farmers are the major contributing factors towards resistance development and health issues. For underdeveloped countries like Pakistan a comprehensive and well planned program targeting on alternative pest control method and use of biological agents along with insecticides need to be initiated that can reduce the total dependency on chemicals.

There is a dier need to start up the farmer's awareness campaigns as well as need to organize farmers by giving them trainings about proper disposal of empty containers to minimize the resistance development and resurgence of secondary pests.

## Supporting information

**S1 Table. Questionnaire.**
(PDF)

**S2 Table. Supplementary data.**
(PDF)

## Acknowledgments

The author sincerely thanks to the farmers and extension agriculture department for participating and contributing to this study. Special thanks to Dr Fayyaz Ahmed (Statistics department), University of Gujrat for his kind help in statistical analysis.

## Author Contributions

**Conceptualization:** Bakhtawer.

**Data curation:** Bakhtawer.

**Supervision:** Sumera Afsheen.

**Writing – original draft:** Bakhtawer.

**Writing – review & editing:** Bakhtawer, Sumera Afsheen.

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
