## [Decision Letter · Decision Letter 0]

30 Jul 2020

PONE-D-20-16862

A Cross Sectional Survey of Knowledge, Attitude and Practices Related to the Use of Insecticides among Farmers in Industrial Triangle of Punjab, Pakistan

PLOS ONE

Dear Dr. Nasrullah,

Thank you for submitting your manuscript to PLOS ONE. After careful consideration, we feel that it has merit but does not fully meet PLOS ONE’s publication criteria as it currently stands. Therefore, we invite you to submit a revised version of the manuscript that addresses the points raised during the review process.

We look forward to receiving your revised manuscript.

Kind regards,

Ahmet Uludag, Ph.D.

Academic Editor

PLOS ONE

Journal Requirements:

3. Please note that maps produced using Google Maps imagery are not compatible with the licence used by PLOS (https://journals.plos.org/plosone/s/licenses-and-copyright). Please replace the map in Fig 1 using an alternative source of imagery that is compatible with our licence. For more information, please see: https://journals.plos.org/plosone/s/figures#loc-maps.

4. We suggest you thoroughly copyedit your manuscript for language usage, spelling, and grammar. If you do not know anyone who can help you do this, you may wish to consider employing a professional scientific editing service.  

7. We note you have included a table to which you do not refer in the text of your manuscript. Please ensure that you refer to Table 5 in your text; if accepted, production will need this reference to link the reader to the Table.

Additional Editor Comments (if provided):

Normally, I reject such papers after one rejection and one major revision that has more criticism. However, I would like to give you another chance to improve your manuscript. Please see both reviewers suggestions in the form and santicised copies of the manuscript. Please improve your language. Statistical analysis is must. You can run multiple correlation analysis or another one (see Maclaren et al 2018). If I continue as editor in the second stage, I will look for all main points suggested by reviewers.

Reviewers' comments:

Reviewer's Responses to Questions

**Comments to the Author**

1. Is the manuscript technically sound, and do the data support the conclusions?

Reviewer #1: No

Reviewer #2: Partly

2. Has the statistical analysis been performed appropriately and rigorously? 

Reviewer #1: No

Reviewer #2: No

3. Have the authors made all data underlying the findings in their manuscript fully available?

Reviewer #1: Yes

Reviewer #2: Yes

4. Is the manuscript presented in an intelligible fashion and written in standard English?

Reviewer #1: No

Reviewer #2: No

5. Review Comments to the Author

Reviewer #1: the introduction section is too short and did not cover the state of the arts in the last 10 years. the methodology section is not well designed and lack of inclusion and exclusion criteria. the statistical analysis is not well done. results section needs to be reformatted. Discussion section is rather weak, did not explain the results, and not connect them with previous published work. the references in this section should be rewritten. the conclusion section is more general and did not show filled gap of knowledge, the reference section is not up-to-date.

Reviewer #2: - The introduction is inadequate and should be expand it. The detailed aim should be given at last paragraph of introduction.

-The reasons for the preference of three sub-regions should be explained and if the differences there are should be given in the text such as crop, irrigation, topographic, climate, etc.

-The abstract mentioned Anova analyses and the result has the univariate regression model. However, the detailed analysis method is not in the material and method. Please give a detailed analysis method.

-There are no statistical results and this decreases its value for publication in this journal.

-I did not understand the terminology (%age) in your tables. You mean percentage, please change it

- Discussion is inadequate please expand it. You have three sub-region, you can compare them

-Reference should be rearranged according to journal guidelines

6. PLOS authors have the option to publish the peer review history of their article (what does this mean?). If published, this will include your full peer review and any attached files.

Reviewer #1: **Yes: **Yasser El-Nahhal

Reviewer #2: No

---

## [Decision Letter · Decision Letter 1]

26 Mar 2021

PONE-D-20-16862R1

A Cross Sectional Survey of Knowledge, Attitude and Practices Related to the Use of Insecticides among Farmers in Industrial Triangle of Punjab, Pakistan

PLOS ONE

Dear Dr. Nasrullahh,

Thank you for submitting your manuscript to PLOS ONE. After careful consideration, we feel that it has merit but does not fully meet PLOS ONE’s publication criteria as it currently stands. Therefore, we invite you to submit a revised version of the manuscript that addresses the points raised during the review process.

Again we will ask major revision. Normally I would have rejected. Could you please follow all points from the first and second rounds. You write a rebuttal point by point explaining why you did not follow reviewer's specific point. If you done it, mention it.

We look forward to receiving your revised manuscript.

Kind regards,

Ahmet Uludag, Ph.D.

Academic Editor

PLOS ONE

Additional Editor Comments (if provided):

Again we will ask major revision. Normally I would have rejected. Could you please follow all points from the first and second rounds. You write a rebuttal point by point explaining why you did not follow reviewer's specific point. If you done it, mention it.

Reviewers' comments:

Reviewer's Responses to Questions

**Comments to the Author**

1. If the authors have adequately addressed your comments raised in a previous round of review and you feel that this manuscript is now acceptable for publication, you may indicate that here to bypass the “Comments to the Author” section, enter your conflict of interest statement in the “Confidential to Editor” section, and submit your "Accept" recommendation.

Reviewer #1: (No Response)

Reviewer #3: (No Response)

2. Is the manuscript technically sound, and do the data support the conclusions?

Reviewer #1: No

Reviewer #3: Yes

3. Has the statistical analysis been performed appropriately and rigorously? 

Reviewer #1: Yes

Reviewer #3: N/A

4. Have the authors made all data underlying the findings in their manuscript fully available?

Reviewer #1: Yes

Reviewer #3: Yes

5. Is the manuscript presented in an intelligible fashion and written in standard English?

Reviewer #1: No

Reviewer #3: Yes

6. Review Comments to the Author

Reviewer #1: the manuscript is not improved. the previous commitments must be considered to timprove the quality of the manuscript.

Reviewer #3: The manuscript was adjusted according to reviewers’ comments; however, it still has limits and should not be published in the submitted version. Suggested revisions can be found directly in the text (pdf). There are many grammatical errors in the text (including errors in active ingredients names), as well as punctuation errors. There is no uniformity in the use of initial capital and small letters in the text. Due to these limitations, the overall impression is then reduced. Chapters could be better structured.

Concerning guidelines of this journal for Figures and Tables, these are not followed: figure captions must be inserted in the text of the manuscript, immediately following the paragraph in which the figure is first cited. Also, the Figure should be abbreviated to “Fig” (e.g. Fig 1, Fig 2, Fig 3, etc).

The other guideline which is not followed is related to References: these should be listed at the end of the manuscript and numbered in the order that they appear in the text. In the text, cite the reference number in square brackets (e.g., “We used the techniques developed by our colleagues [19] to analyze the data”).

7. PLOS authors have the option to publish the peer review history of their article (what does this mean?). If published, this will include your full peer review and any attached files.

Reviewer #1: No

Reviewer #3: No

---

## [Decision Letter · Decision Letter 2]

16 Jun 2021

PONE-D-20-16862R2

A Cross Sectional Survey of Knowledge, Attitude and Practices Related to the Use of Insecticides among Farmers in Industrial Triangle of Punjab, Pakistan

PLOS ONE

Dear Dr. Bakhtawer,

Thank you for submitting your manuscript to PLOS ONE. After careful consideration, we feel that it has merit but does not fully meet PLOS ONE’s publication criteria as it currently stands. Therefore, we invite you to submit a revised version of the manuscript that addresses the points raised during the review process.

There are some editorial, lingual and presenting problems. I hope you can improve all quickly.

We look forward to receiving your revised manuscript.

Kind regards,

Ahmet Uludag, Ph.D.

Academic Editor

PLOS ONE

Journal Requirements:

Additional Editor Comments:

I have been following your paper due to being editor. I think, you need help from a native English speaker to improve the language in the text although it can be understandable for some colleagues. I also recommend that change some tables to basic graphs. It can be more visual and understandable for the audience. Actually you have almost done with this paper. Please check grammatical and editorial mistakes carefully.

Reviewers' comments:

Reviewer's Responses to Questions

**Comments to the Author**

1. If the authors have adequately addressed your comments raised in a previous round of review and you feel that this manuscript is now acceptable for publication, you may indicate that here to bypass the “Comments to the Author” section, enter your conflict of interest statement in the “Confidential to Editor” section, and submit your "Accept" recommendation.

Reviewer #1: All comments have been addressed

Reviewer #3: (No Response)

2. Is the manuscript technically sound, and do the data support the conclusions?

Reviewer #1: Yes

Reviewer #3: Partly

3. Has the statistical analysis been performed appropriately and rigorously? 

Reviewer #1: Yes

Reviewer #3: Yes

4. Have the authors made all data underlying the findings in their manuscript fully available?

Reviewer #1: Yes

Reviewer #3: Yes

5. Is the manuscript presented in an intelligible fashion and written in standard English?

Reviewer #1: Yes

Reviewer #3: Yes

6. Review Comments to the Author

Reviewer #1: the manuscript is improved

Reviewer #3: (No Response)

7. PLOS authors have the option to publish the peer review history of their article (what does this mean?). If published, this will include your full peer review and any attached files.

Reviewer #1: No

Reviewer #3: No

---

## [Author Response · Author response to Decision Letter 2]

2 Jul 2021

Already provided in separate file

---

## [Decision Letter · Decision Letter 3]

19 Jul 2021

A Cross Sectional Survey of Knowledge, Attitude and Practices Related to the Use of Insecticides among Farmers in Industrial Triangle of Punjab, Pakistan

PONE-D-20-16862R3

Dear Dr. nasrullah,

We’re pleased to inform you that your manuscript has been judged scientifically suitable for publication and will be formally accepted for publication once it meets all outstanding technical requirements.

Kind regards,

Ahmet Uludag, Ph.D.

Academic Editor

PLOS ONE

Additional Editor Comments (optional):

Dear Author, please go through text and make it more precise for language.

Reviewers' comments:

Reviewer's Responses to Questions

**Comments to the Author**

1. If the authors have adequately addressed your comments raised in a previous round of review and you feel that this manuscript is now acceptable for publication, you may indicate that here to bypass the “Comments to the Author” section, enter your conflict of interest statement in the “Confidential to Editor” section, and submit your "Accept" recommendation.

Reviewer #4: All comments have been addressed

2. Is the manuscript technically sound, and do the data support the conclusions?

Reviewer #4: Yes

3. Has the statistical analysis been performed appropriately and rigorously? 

Reviewer #4: Yes

4. Have the authors made all data underlying the findings in their manuscript fully available?

Reviewer #4: Yes

5. Is the manuscript presented in an intelligible fashion and written in standard English?

Reviewer #4: Yes

6. Review Comments to the Author

Reviewer #4: (No Response)

7. PLOS authors have the option to publish the peer review history of their article (what does this mean?). If published, this will include your full peer review and any attached files.

Reviewer #4: No

---

## [Editor Report · Acceptance letter]

29 Jul 2021

PONE-D-20-16862R3 

A Cross Sectional Survey of Knowledge, Attitude and Practices Related to the Use of Insecticides among farmers in Industrial Triangle of Punjab, Pakistan. 

Dear Dr. Nasrullah:

I'm pleased to inform you that your manuscript has been deemed suitable for publication in PLOS ONE. Congratulations! Your manuscript is now with our production department. 

Kind regards, 

on behalf of

Dr. Ahmet Uludag 

Academic Editor

PLOS ONE